# Whole Exome Sequencing Identifies PHF14 Mutations in Neurocytoma and Predicts Responsivity to the PDGFR Inhibitor Sunitinib

**DOI:** 10.3390/biomedicines10112842

**Published:** 2022-11-08

**Authors:** Dongyun Zhang, William H. Yong, Masoud Movassaghi, Fausto J. Rodriguez, Issac Yang, Paul McKeever, Jiang Qian, Jian Yi Li, Qinwen Mao, Kathy L. Newell, Richard M. Green, Cynthia T. Welsh, Anthony P. Heaney

**Affiliations:** 1Department of Medicine, David Geffen School of Medicine, University of California, Los Angeles, CA 90095, USA; 2Department of Pathology and Laboratory Medicine, David Geffen School of Medicine, University of California, Los Angeles, CA 90095, USA; 3Department of Neurosurgery, David Geffen School of Medicine, University of California, Los Angeles, CA 90095, USA; 4Department of Pathology and Clinical Laboratories, University of Michigan, Ann Arbor, MI 48109, USA; 5Department of Pathology, Albany Medical Center, Albany, NY 12208, USA; 6Department of Pathology and Laboratory Medicine, North Shore University Hospital and Long Island Jewish Medical Center, Manhasset, NY 11040, USA; 7Donald and Barbara Zucker School of Medicine at Hofstra/Northwell, Northwell Health System, Lake Success, NY 11549, USA; 8Department of Pathology, University of Utah, Salt Lake City, UT 84112, USA; 9Department of Pathology and Laboratory Medicine, Indiana University School of Medicine, Indianapolis, IN 46202, USA; 10Neuro-Oncology Program, Kaiser Los Angeles Medical Center, Los Angeles, CA 90027, USA; 11Department of Pathology and Laboratory Medicine, Medical University of South Carolina, Charleston, SC 29425, USA

**Keywords:** neurocytoma, plant homeodomain finger protein 14 (PHF14), platelet derived growth factor receptor-alpha (PDGFRα), Sunitinib, whole exome sequencing (WES)

## Abstract

Neurocytomas are rare low-grade brain tumors predominantly affecting young adults, but their cellular origin and molecular pathogenesis is largely unknown. We previously reported a sellar neurocytoma that secreted excess arginine vasopressin causing syndrome of inappropriate anti-diuretic hormone (SIADH). Whole exome sequencing in 21 neurocytoma tumor tissues identified somatic mutations in the *plant homeodomain finger protein 14 (PHF14)* in 3/21 (14%) tumors. Of these mutations, two were missense mutations and 4 caused splicing site losses, resulting in PHF14 dysfunction. Employing shRNA-mediated knockdown and CRISPR/Cas9-based knockout approaches, we demonstrated that loss of PHF14 increased proliferation and colony formation in five different human, mouse and rat mesenchymal and differentiated cell lines. Additionally, we demonstrated that PHF14 depletion resulted in upregulation of platelet derived growth factor receptor-alpha (PDGFRα) mRNA and protein in neuroblastoma SHSY-5Y cells and led to increased sensitivity to treatment with the PDGFR inhibitor Sunitinib. Furthermore, in a neurocytoma primary culture harboring splicing loss PHF14 mutations, overexpression of wild-type PHF14 and sunitinib treatment inhibited cell proliferation. Nude mice, inoculated with PHF14 knockout SHSY-5Y cells developed earlier and larger tumors than control cell-inoculated mice and Sunitinib administration caused greater tumor suppression in mice harboring PHF-14 knockout than control SHSY-5Y cells. Altogether our studies identified mutations of PHF14 in 14% of neurocytomas, demonstrate it can serve as an alternative pathway for certain cancerous behavior, and suggest a potential role for Sunitinib treatment in some patients with residual/recurrent neurocytoma.

## 1. Introduction

Neurocytomas are rare, represent 0.1–0.5% of primary intracranial tumors, and typically affect young adults [1]. Neurocytomas are categorized based on their location, namely central neurocytomas (CN) which involve the septum pellucidum, fornix, or lateral ventricular walls; whereas extraventricular neurocytomas (EVN) occur in the frontal or temporal lobes, spinal cord brain stem, and rarely the sellar region [2,3,4]. Defined as Grade II by the latest world health organization (WHO) classification, neurocytomas usually exhibit a benign course with good outcome if gross-total surgical resection (GTR) is achieved. However, subtotal tumor resection (STR) portends a less favorable prognosis due to increased recurrence and these tumors can exhibit high proliferative activity and prominent vascular proliferation, resulting in decreased survival rates [5,6]. Although stereotactic radiosurgery (SRS) and conventional radiotherapy can be effective to manage residual and recurrent tumors, radiation-induced side effects such as short-term memory impairment and other late neurotoxicity have been observed in more than half of treated patients [7]. 

Although genomic studies in neurocytomas are limited, comparative genomic hybridization analysis in 20 neurocytomas revealed frequent gains mapped to 2p24.1-22.1, 10q23.3-26.3, 11q23-25, and 18q21.3-qter, and frequent losses at 1pter-36.3, 1p34.3, 6q13-21, 12q23-qter, 17p13.3, 17q11-23 and 20pter-12.3 [8]. SNP array analyses further demonstrated mosaic gains of 1p, 2q, 6q, 12p, 20q and losses at 19p [9]. DNA methylation analysis in 40 EVN demonstrated that 22 of the 40 cases (55%) formed a separate epigenetic group from the other samples and copy number analysis revealed half of these cases harbored gene fusion of FGFR1-TACC1 (*n* = 11) or FGFR3-TACC3 (*n* = 2) [10]. FGFR-TACC rearrangement facilitates FGFR dimerization, autophosphorylation and constitutive FGFR tyrosine kinase activation, leading to increased cell proliferation and disease progression [10]. Additionally, an EWSR1-ATF1 fusion and a MUTYH G382D mutation were reported following next generation sequencing in a pediatric periventricular atypical central neurocytoma [11]. 

We recently described a sellar neurocytoma that secreted excess arginine vasopressin which led to presentation with severe hyponatremia [4]. To better understand the role of genetic mutations in neurocytoma, we performed whole exome sequencing on 21 samples. Multiple somatic mutations in the plant homeodomain finger protein 14 (PHF14) were identified in 3/21 (14%) neurocytomas. Two of these were missense and 4 were at the splice sites of intronic regions. PHF14 belongs to the plant homeodomain (PHD) finger protein superfamily, a group of evolutionary conserved histone binding partners which control gene expression by orchestrating multiple complexes of chromatin regulators and transcription factors in a histone sequence- and state-specific manner [12]. Canonical PHD finger is a small zinc-binding motif (about 50–80 amino acids) characterized by a Cys4-His-Cys3 architecture, and adopts a stable two-strand anti-parallel β-sheet and one or two C-terminal α-helix structures coordinated by two zinc atoms [13]. Additionally, some members of this family include a non-canonical extended PHD (ePHD) finger characterized as Cys2-His-Cys5-His-Cys2-His structure [14]. PHF14 belongs to this sub-family and contains two canonical PHD fingers (aa 321–377 and aa 727–776) and one non-canonical ePHD finger (aa 385–498). Animal studies demonstrate PHF14 plays an important role in development of multiple organs during embryogenesis [15,16]. Homozygous PHF14 knockout mice died within hours after birth due to respiratory failure, although heterozygous PHF14 knockout animals are healthy and fertile [15,16]. Abnormal pulmonary alveolus wall hypertrophy and upregulation of platelet derived growth factor-alpha (PDGFRα) expression was seen in the absence of PHF14 and resulted in uncontrolled mesenchymal cell proliferation and interstitial hyperplasia [16], suggesting a key epigenetic regulatory role for PHF14 in organogenesis.

In additional studies, we used shRNA-directed knockdown and CRISPR/Cas9-based knockout approaches in multiple human, mouse and rat cell-lines to characterize how loss of PHF14 action might affect neuronal proliferation and PDGFR expression and potentially contribute to development and growth of neurocytomas and demonstrated that Sunitinib effectively inhibited growth of a neurocytoma primary culture. Altogether our studies demonstrate a significant mutation frequency of PHF14 in neurocytomas and predict a favorable response to Sunitinib treatment in patients with neurocytomas harboring a PHF14 defect. 

## 2. Results

### 2.1. PHF14 Mutations in Human Neurocytoma

Neurocytoma was first described by Hassoun et al. in 1982 [17] and ~500 cases have been reported world-wide [18]. To obtain sufficient neurocytoma tissues for our studies, we collaborated with 11 centers in USA and Canada to gather 54 neurocytoma formalin-fixed, paraffin-embedded (FFPE) and frozen tissues. Pathological re-evaluation was conducted in all cases (by W. Y.) to confirm the diagnosis of neurocytoma. Of the 54 tissues, 38 had a tumor area >95% and diameter >5 mm suitable for genomic DNA extraction, but only 21 tumor tissues (FFPE samples *n* = 17, frozen samples *n* = 4) contained sufficient DNA mass and integrity for whole exome sequencing (WES, Figure 1A). Eighteen out of the twenty-one samples were central neurocytomas (85.7%), among which 3 samples were from recurrent diseases (14.3%, Figure 1B). The remainder comprised one extraventricular neurocytoma (4.8%), but no information was available on the tumor location for the other two samples (Figure 1B). Autopsy cerebellum FFPE samples were used as controls (*n* = 5). The observed PHF14 variants occurring at two exonic sites, G476S (in sample #N27het), and G321V (in sample #N26daub) and four intronic sites c. 901-3T>G, c.1980+1G>C, c.1980+4A>T, and c.2654+1G>A (in sample #N36dip), have not been reported previously (Figure 1C). 

### 2.2. PHF14 Knockdown Enhances Cell Proliferation and Increases Anchorage Independent Cell Growth

PHF14 (also known as KIAA0783) is highly conserved from *C. elegans* to humans, sharing 97.3% and 96.9% protein sequence identity between human, mouse, and rat, respectively [15]. To investigate the biological functions of PHF14, we used shRNA to target a totally conserved 19-bp region in human, rat and mouse PHF14 Exon 10 (CGC ATG ATT CAA ATT CAG GAA) (Figure 2A, Top Panel). Five different cell lines of human, mouse and rat mesenchymal and differentiated cell origin were used to evaluate broadly the effects of shRNA PHF14 knockdown (Figure 2A, Middle Panel). PHF14 knockdown in Puromycin-selected stable transfectants was confirmed by Western Blot and quantified by densitometric analysis as demonstrated in Figure 2A (Bottom Panel). As depicted in Figure 2B, depletion of PHF14 resulted in increased cell proliferation in all the cell lines, indicating that loss of PHF14 broadly promotes cell growth in both mesenchymal and differentiated cells. As no neurocytoma cell lines exist, we used neuroblastoma SHSY-5Y cells as a surrogate model to examine effects of PHF14 loss on anchorage-independent cell growth, an indicator of tumor cell aggressiveness and metastatic potential. Neuroblastomas are derived from migratory neural crest and manifest morphological and biochemical features of immature sympathoblasts [19,20]. shRNA PHF14 SHSY-5Y cells formed larger colonies than those observed in nonsense-transfected control cells (Colony number with radius > 100 μm: shRNA PHF14, 413 ± 72 vs. Nonsense, 118 ± 72, *p* < 0.05, Figure 2C). We next used CRISPR with a single guiding RNA (sgRNA) to target a 20-bp human PHF14 exon 1 region (TGG ATC GCA GCT CCA AGA GG) to knockout PHF14 (Figure 2D, Left Panel). Similar to what we had observed in PHF14 knockdown transfectants, PHF14 knockout SHSY-5Y cells exhibited higher cell proliferation (Figure 2D, Middle Panel) and increased colony size in soft agar (Colony number with radius > 100 μm; sgRNA, 472 ± 118 vs. Control, 118 ± 72, *p* < 0.05, Figure 2D, Right Panel). Altogether, our results demonstrate that loss of PHF14 results in increased cell proliferation and anchorage-independent growth and may serve as an alternative pathway for certain cancerous behavior.

### 2.3. PHF14 Depletion Increases PDGFRα Expression and Responsivity to Sunitinib

To understand how loss of PHF14 upregulates cell proliferation, we compared the transcriptome in SHSY-5Y nonsense control and shRNA PHF14 knock-down cells using RNA microarray. Appendix A showed genes with expression changes more than 1.5-fold following PHF14 knockdown. Gene Ontology enrichment analysis revealed changes in proteinaceous extracellular matrix organization (Appendix A). The platelet derived growth factor receptors (PDGFRs) are a group of cell surface tyrosine kinase receptors for the platelet-derived growth factor (PDGF) family, which are mitogenic for cells of mesenchymal origin [21]. Upon ligand binding, PDGFRs undergo dimerization and autophosphorylation, and activate several downstream signaling pathways involved in organ development, wound healing, and tumor progression [22]. PHF14 has been implicated in trans-repression of PDGFRα in mesenchymal cells possibly by interaction with other leucine zipper domain containing transcription factors, although no direct interaction between PHF14 and PDGFRα promoter has been confirmed [16]. shRNA PHF14 knockdown in differentiated SHSY-5Y and mesenchymal NIH3T3 cells led to increased PDGFRα mRNA (Figure 3A) and protein (Figure 3B) expression compared to nonsense shRNA cells. Increased PDGFRα expression was confirmed in stable CRISPR PHF14 knockout SHSY-5Y cells (Figure 3C), supporting the inhibitory role of PHF14 in regulation of PDGFRα expression. Several potent PDGFR inhibitors such as Imatinib and Sunitinib are now clinically approved to treat leukemia [23], gastrointestinal stromal tumor [24], renal cell carcinoma [25] and pancreatic neuroendocrine tumors [26]. Given the changes in PDGFR expression we had observed in PHF14 knock-down or knock-out SHSY-5Y cells, we next compared the effects of Imatinib and Sunitinib on cell proliferation in SHSY-5Y cells. Firstly, we observed that SHSY-5Y cells were more sensitive to Sunitinib-induced growth inhibition compared to Imatinib treatment (Figure 3D). Additionally, we demonstrated that SHSY-5Y shRNA PHF14 cells were relatively resistant to Imatinib-induced growth inhibition (Figure 3E), and more sensitive to Sunitinib treatment (Figure 3F) compared with nonsense controls. This was confirmed in PHF14 knockout SHSY-5Y cells which also showed greater sensitization to Sunitinib (Figure 3G). In summary, we demonstrate that loss of PHF14 increases SHSY-5Y PDGFRα expression and increases their responsiveness to the tyrosine kinase inhibitor (TKI), Sunitinib.

### 2.4. Overexpression of PHF14 Inhibits the Growth of Neurocytoma Primary Culture with PHF14 Defect

Neurocytomas are typically well-differentiated neuronal tumors characterized by the presence of synaptic structures, dense core vesicles, and parallel microtubules [27]. They express synaptophysin (SYP), neuronal nuclear antigen (NeuN), and neurofilament (NF) but glial markers such as glial fibrillary acidic protein (GFAP), vimentin (VIM) or S-100, are usually absent [28]. However, transition in cell morphology and phenotype has been observed during in vitro primary neurocytoma culture [27,28], pointing to their possible neuronal stem cell origin. To characterize the potential impact of in vitro culture conditions to select certain cell populations and identify the in vitro culture conditions that maintain the neuronal phenotype of neurocytomas, we compared neurocytoma cell morphology using two distinct media. Insulin, transferrin, selenium and fibronectin (ITSFn) media has been reported to preserve a neuronal phenotype whereas ITSFn media supplemented with 5% FBS reportedly favors astrocyte outgrowth [27,28]. We observed that primary neurocytoma cells formed aggregates or neurospheres in the ITSFn media that were loosely attached to the plate surface and displayed a rounded cell morphology with prominent cytoplasmic processes (Figure 4A, Upper Panel). In contrast, neurocytoma cells cultured in ITSFn with 5% FBS became elongated with multipolar cytoplasmic sprouting, reflecting a more astrocytic-like cell shape (Figure 4A, Bottom Panel). We therefore used ITSFn media without serum to maintain a neuronal phenotype in the neurocytoma primary culture. Interestingly, whole exome sequencing of this neurocytoma revealed 4 variants in PHF14 at intronic regions c.901-3T>G (NM_014660 intron 3), c.1980+1G>C/+4A>T (intron 9), and c.2654+1G>A (intron 13, Figure 4B). As these splice site losses resulted in disrupted PHF14 function, we used lentivirus infection to introduce wild type full length PHF14 into the primary neurocytoma cells (Figure 4C). Overexpression of wild type PHF14 (Relative PHF14 mRNA expression: Vector 1.1 ± 0.2, PHF14 276 ± 12, *p* < 0.001, Figure 4C) reduced cell viability of the primary neurocytoma culture (Cell Viability, Vector, 1.0 ± 0.08, PHF14 0.78 ± 0.01, *p* < 0.05, Figure 4C) supporting our hypothesis that PHF14 mutant contributed to increased cell proliferation in the neurocytoma. We next evaluated sensitivity of the primary culture to several pathway inhibitors, including the PDGFR inhibitor Sunitinib, the MAPK/ERK inhibitor MEK-162, and the AMPK/AKT inhibitor Metformin. Consistent with what we observed previously in the PHF14 knockdown neuroblastoma SHSY-5Y cells, the primary neurocytoma cells exhibited greater sensitivity to Sunitinib treatment (5 μM, 0.1 ± 0.01 vs. 1.0 ± 0.04, *p* < 0.005) when compared to the MAPK/ERK or AMPK/AKT inhibitors MEK-162 and Metformin (Figure 4D). Our findings suggest that Sunitinib could be a feasible treatment choice for some neurocytoma patients that exhibit PHF14 defects.

### 2.5. Confirmation of the Tumor Inhibitory Effect of PHF14 and Sunitinib In Vivo

To characterize effects of PHF14 depletion on tumor growth *in vivo*, we inoculated PHF14 knockout (sgRNA-PHF14) SHSY-5Y cells and SHSY-5Y control cells subcutaneously into five-week-old male athymic nude mice (Nu/J, *n* = 10) at the density of 2 × 10^6^ cells/animal. Mice were examined for tumor presence twice weekly, and tumor volume was measured following palpable tumor development. Mice inoculated with SH-SY5Y sgRNA-PHF14 developed tumors earlier (2 weeks) than mice inoculated with control neuroblastoma cells (4.5 weeks) (Figure 5A). Animals were euthanized at 5 weeks due to tumor size at which time tumor volumes were higher in SH-SY5Y sgRNA-PHF14 inoculated mice compared to controls [tumor weight (g); control, 0.467 ± 0.39 versus sgRNA-PHF14, 1.5 ± 0.21, *p* < 0.0005, Figure 5B], confirming the tumor inhibitory effect of PHF14 *in vivo*. Furthermore, administration of Sunitinib (40 or 80 mg/kg) by oral gavage profoundly inhibited tumor growth (Figure 5C) with greater tumor suppression observed in mice harboring the PHF-14 knockout sgRNA cells (tumor weight (g); control, Vehicle: 0.32 ± 0.27; Sunitinib 40 mg/kg: 0.06 ± 0.029, *p* < 0.01; Sunitinib 80 mg/kg: 0.071 ± 0.039, *p* < 0.01 versus sgRNA-PHF14, Vehicle: 0.61 ± 0.45; Sunitinib 40 mg/kg: 0.083 ± 0.071 *p* < 0.05, Sunitinib 80 mg/kg: 0.041 ± 0.043, *p* < 0.05, Figure 5D).

## 3. Discussion

Neurocytomas are generally benign slow growing brain tumors, and depending on tumor localization, present with symptoms of blurred vision, headache, nausea, vomiting and seizures due to raised intracranial pressure and obstructive hydrocephalus [6]. CT and MRI scans localize the tumors but the ultimate diagnosis largely depends on histopathological analysis of resected tissue [29] as distinguishing neurocytomas from other types of brain tumors, such as oligodendrogliomas, clear cell ependymomas, pineocytoma, or neuroblastomas, can be challenging. Neurocytomas classically exhibit a honeycomb-like arrangement, small round or oval nuclei, scant cytoplasm, and a perinuclear halo, sometimes resembling the pattern of oligodendrogliomas; but they also contain large fibrillary areas resembling pineocytoma; with cells arranged in a straight-line and pseudorossettes resembling those of clear cell ependymomas in some cases [29]. Additional molecular analyses are needed to better distinguish neurocytomas from other tumor types to guide appropriate therapy and prognosis [30,31]. 

Next generation sequencing (NGS) has provided an unbiased high throughput approach to investigate genomic features of tumors to identify novel diagnostic biomarkers that may predict tumor behavior or potentially actionable mutations for disease treatment and management [32]. Herein we describe mutations in PHF14 gene in a series of neurocytomas. Disruptive PHF14 mutations were found in 3 of 21 (14%) neurocytoma tissues by WES. Our additional studies of PHF14 function in several cell lines corroborated by a single neurocytoma primary culture point to a putative regulatory role of PHF14 in modulating cell proliferation, PDGFRα expression and sensitivity to the anti-proliferative effect of the PDGFR inhibitor Sunitinib. Our observed in vitro effects were further confirmed in a xenograft animal model using neuroblastoma SHSY-5Y cells. Our studies indicate that lack of PHF14 may be a predictive biomarker for Sunitinib treatment in neurocytomas.

Point mutations, deletion or chromosomal translocations in the PHF14 plant homeodomain family fingers are found in a wide range of pathologies, including cancer, mental retardation and immunodeficiency [33]. PHF14 homozygous deletion has been identified in the human biliary tract cancer cell line OZ [34] and a bi-allelic inactivating mutation is present in human colon cancer HCT-116 cells [35], suggesting aberrant PHF14 expression may play a role in several tumor types. Conversely, ~ 70% of non-small lung cancer tissues exhibited PHF14 overexpression and was significantly associated with reduced patient survival [36].

Animal studies show that PHF14 controls mesenchymal cell proliferation by directly repressing PDGFRα expression [15,16]. In a ligand-dependent manner, PDGFRα forms homo- or hetero-dimers with PDGFRβ, and phosphorylates PIK3R1, the regulatory subunit of p110/p85 complex to activate the PI3K/AKT pathway. Ligand bound PDGFR also activates PLCγ1, to induce the secondary messengers, diacylglycerol, inositol 1,4,5-trisphosphate, and cytosolic Ca^2+^, leading to activation of protein kinase C [21]. Mutations in PDGFRα have been associated with several cancers including somatic and familial gastrointestinal stromal tumors, in addition to idiopathic hypereosinophilic syndrome. Small molecule multi-target tyrosine kinase inhibitors (TKI) such as Sunitinib profoundly inhibit PDGFRα/β activation. It also inhibits several other receptors including vascular endothelial cell growth factor (VEGF) receptors (VEGFR1-VEGFR3), stem cell receptor (Kit), Fms-like tyrosine kinase receptor-3 (FLT-3), colony stimulating factor receptor (CSF-1R), and glial cell line-derived neurotrophic factor receptor (RET) [22]. Although its lack of specificity potentially contributes to side effects, it has shown efficacy in non-small cell lung, renal cell, breast, gastrointestinal stromal and neuroendocrine cancers [22]. Common adverse effects of Sunitinib are typically mild and include fatigue, diarrhea, neutropenia, leukopenia, lymphopenia and thrombocytopenia.

Our in vitro studies using a neurocytoma primary culture harboring mutations resulting in loss of PHF14 function demonstrated that restoration of PHF14 function by wild-type PHF14 transfection led to reduced primary neurocytoma cell viability and the PHF14 neurocytoma cell transfectants exhibited greater sensitivity to Sunitinib treatment.

In summary, we have identified a genetic variant of PHF14 in 14% of neurocytomas using WES and demonstrated that loss of PHF14 function resulted in increased cell proliferation in several cell lines and in a neurocytoma primary culture. PHF14 depletion enhanced PDGFRα expression and increased proliferative inhibitory responses to Sunitinib treatment. Altogether our studies support a putative tumor proliferation inhibition role of PHF14 in some neurocytomas, and suggest that Sunitinib could be a feasible treatment choice for some patients with residual/recurrent neurocytoma.

## 4. Materials and Methods

### 4.1. DNA Isolation and Whole Exome Sequencing

Genomic DNA (gDNA) isolation using QIAamp DNA FFPE Tissue Kit (Qiagen, Germantown, MD, USA). Quality of gDNA was checked using the Agilent Test Genomic DNA Analysis (Agilent, Santa Clara, CA, SUA), and gDNA quantity was measured using Nanodrop and Qubit fluorometer 3.0 (Thermo Fisher Scientific, Waltham, MA, USA). The Nextera rapid capture was used for library preparation and exome enrichment (Illumina Inc., San Diego, CA). Sequencing was performed at the UCLA Neuroscience Genomic Core facility (University of California, Los Angeles, United States) using “MiSeq” Next generation sequencing system (Illumina Inc., San Diego, CA, USA) with paired-end (PE) 2 × 100 bp.

### 4.2. Bioinformatic Analysis of WES Data

The sequence data were aligned to the GRCh38 human reference genome using Partek_flow bwa-0.7.17 (bwa mem). PCR duplicates were marked using Mark Duplicates program in picard_tools/2.13.2 tool set. GATK v4.0.8.1 was used for insertions and deletions (INDEL) realignment and base quality recalibration. GATK v4.0.8.1 Mutect2 was used to call the single nucleotide variants (SNVs) and small INDELs between normal vs. tumor pairs. All variants were annotated using the WAnnovar program.

### 4.3. Cell Culture and Reagents

SHSY-5Y human neuroblastoma cells were purchased from ATCC (CRL-2266) and cultured as a monolayer at 37 °C, 5% CO_2_ in Dulbecco’s Modified Eagles Medium (DMEM)/F12 containing 10% fetal bovine serum (FBS), and penicillin/streptomycin. HEK293, NIH3T3, AtT20 and PC-12 were cultured in DMEM with 10% FBS, all reagents from Life Technologies, Inc (Grand Island, NY, USA). All cell cultures were detached with trypsin and transferred to new 75 cm^2^ culture flasks (Fisher, Pittsburgh, PA, USA) once weekly. Human recombinant bFGF, EGF, Metformin, MEK-162, Imatinib and Sunitinib were purchased from Sigma-Aldrich Corp. (St. Louis, MO, USA). The growth factors and Metformin were dissolved in media while the other compounds were dissolved in DMSO. Small aliquots were stored at −80 °C and used after a single freeze–thaw cycle.

### 4.4. RNA Microarray Processing and Bioinformatic Analysis

Total RNA was extracted from SHSY-5Y nonsense control cells and shRNA PHF14 knockdown cells using a Qiagen RNeasy Mini kit (Qiagen, Germantown, MD, USA). RNA quantity and quality were assessed using Agilent TapeStation system and amplified to biotin-labelled cDNA using NuGEN cDNA systems (Tecan Group Ltd., Männedorf, Switzerland). Illumina Human HT-12 Expression BeadChip v4 microarray (Agilent, Santa Clara, CA) was used for whole-genome gene expression profiling, and data were analyzed by the UCLA Neuroscience Genomics Core (UNGC) using R and Bioconductor. The gene ontology (GO) enrichment analysis was performed to assess changes led by PHF14 knockdown using a Functional Annotation of NIH DAVID program with Fisher exact *t*-test *p* value < 0.05.

### 4.5. Human Neurocytoma Primary Cultures

Fresh surgically resected human neurocytoma tumor tissues were washed and minced with blades in PBS. After centrifugation, cell pellets were resuspended in either serum free DMEM/F12 (1:1) medium containing insulin, transferrin, selenium and fibronectin (ITSFn) supplemented with 1 × ITS-G (Cat# 41400045, Life Technologies, Inc., Grand Island, NY, USA), 10 ng/mL EGF, and 20 ng/mL bFGF [37], or the differentiation medium which was ITSFn media and 5% FBS (F2442, Sigma-Aldrich Corp., St. Louis, MO, USA). Confluent cells were detached using a soft rubber scraper.

### 4.6. Plasmid Constructs, Transfection and Viral Transduction

ShRNA PHF14 (TRCN0000312505) was purchased from Sigma-Aldrich Corp. sgRNA/Cas9 all-in-one expression clone targeting PHF14 (HCP223067-CG01-3-B) and scrambled sgRNA control were from GeneCopoeia, Inc. (Rockville, MD, USA). All constructs were verified by sequencing. Lipofectamine 2000 transfection reagent was purchased from Invitrogen (Thermo Fisher Scientific Inc., Grand Island, NY, USA). shRNA PHF14 and control stable transfectants were established by puromycin selection (0.5 μg/mL) and the PHF14 sgRNA knockout cells were established by cloning after two-day selection with G418 (1000 μg/mL). PHF14 ORFeome (LPP-OL07464-LX304-050-S) and negative control Lentifect (LPP-NEG-Lv105-025-C) purified lentiviral particles were purchased from GeneCopoeia, Inc (Rockville, MD, USA) at a titer of 2.4 × 10^7^ TU/mL and 6.61 × 10^8^ TU/mL, respectively, using 1 × 10^5^ TU per 1 × 10^6^ neurocytoma primary cells.

### 4.7. Cell Proliferation Assay

shRNA PHF14 and control stable transfectants or neurocytoma primary culture were suspended in 100 μL media, and plated in 96-well plates (2 × 10^3^ viable cells/well) and cultured overnight. Following individual treatment, cell viability was determined using CellTiter-Glo^®^ Luminescent Cell Viability Assay kit (Promega, Madison, WI, USA) with a luminometer (Wallac 1420 Victor 2 multipliable counter system, Ramsey, MN, USA). Results are presented as proliferation index (relative luminescence signal to medium control) and all experiments were repeated in triplicate at least three times with depiction of medium ± standard error (Mean ± SE).

### 4.8. Anchorage-Independent Growth Assay

Anchorage-independent growth (soft agar assay) was performed as described in our previous studies [38]. Briefly, 1 × 10^5^ cells suspended in 0.33% soft agar were seeded over a 0.5% agar layer in 10% FBS DMEM in 6-well plates which were incubated in 5% CO_2_ incubator at 37 °C for 3 weeks. Colonies were inspected under a microscope and only colonies with over ≥32 cells were counted.

### 4.9. Real-Time PCR

Total RNA was extracted with RNeasy kit (Qiagen, Germantown, MD, USA) and RNA quantified and its integrity verified by absorbance measurement at 260 and 280 nm. Total RNA was reverse transcribed into first-strand cDNA using a cDNA synthesis kit (Invitrogen). Quantitative PCR reactions were carried out using CFX Real-time PCR Detection System (Bio-Rad Laboratories Inc., Hercules, CA, USA). Primer sequences were as follows: human *PHF14* forward primer, 5′-GAT AGG TTA GAC AGA AAG TGG AAG-3′; human *PHF14* reverse primer, 5′-CTT TGG CAC GAT ACT GCT GAA GC-3′; human *PDGFRa* forward primer, 5′-AAA GAA GTT CCA GAC CAT CCC-3′; human *PDGFRa* reverse primer, 5′-AGG TGA CCA CAA TCG TTT CC -3′; human *SYP* forward primer, 5′-TCG GCT TTG TGA AGG TGC TGC A-3′; human *ACTB* forward primer, 5′-CAC CAT TGG CAA TGA GCG GTT C-3′, human *ACTB* reverse primer, 5′-AGG TCT TTG CGG ATG TCC ACG T-3′.

### 4.10. Western Blotting

Proteins were extracted in radioimmune precipitation assay (RIPA) buffer (Cell Signaling, Danvers, MA, USA) containing a protease inhibitor cocktail (Roche Molecular Biochemicals, Indianapolis, IN, USA). Protein concentrations were determined by DC protein assay reagent (Bio-Rad, Hercules, CA, USA) and extracts resolved by SDS/PAGE, and transferred to PVDF membranes (Bio-Rad, Hercules, CA, USA). Membranes were then blocked for 2 h at room temperature in 0.1% TBS-Tween-20 containing 5% nonfat dried milk, washed, and then incubated with the specific primary antibodies, anti-PDGFRα (#3164, Cell Signaling Technology, Danvers, MA, USA); Anti-Actin (sc-1616, Santa Cruz Biotechnology Inc., Dallas, TX, USA); anti-PHF14 (HPA000538, Sigma-Aldrich). After washing, membranes were incubated with HRP-conjugated secondary antibodies (Santa Cruz Biotechnology Inc., Dallas, TX, USA) and proteins visualized using a Super Signal Chemiluminescence Assay kit (Pierce, Grand Island, NY, USA). The results shown were representative of three independent experiments.

### 4.11. Tumor Xenograft Model

The use of mice was approved by the University of California Los Angeles (UCLA) Animal Research Committee and complied with all relevant federal guidelines and institutional policies. SH-SY5Y sgRNA-PHF14 or control neuroblastoma cells (2 × 10^6^) in 100 μL matrigel were injected subcutaneously into five-week-old male Nu/J (JAX) mice to generate neuroblastoma tumors (*n* = 10 each group). Tumor presence was checked twice weekly. Tumor diameters were measured in two dimensions with Vernier calipers and volumes calculated using the equation length × width^2^ × 0.5. When the tumor diameter reached 2 cm, mice were euthanized using CO_2_ inhalation. Tumors were excised, and weighed. To determine the effects of Sunitinib treatment on tumor growth in vivo, Sunitinib (dissolved in 0.5% CMC) was administered by oral gavage following SHSY-5Y cells inoculation. Tumor growth was monitored and measured daily, and the animals were euthanized on day 21 after drug treatment due to animal condition, after which tumors were excised and weighed.

### 4.12. Statistics

All in vitro experiments were repeated in triplicate at least three times. Results are expressed as mean ± SE. Differences were assessed student *t* test. *p* values less than 0.05 were considered significant.

## Figures and Tables

**Figure 1 biomedicines-10-02842-f001:**
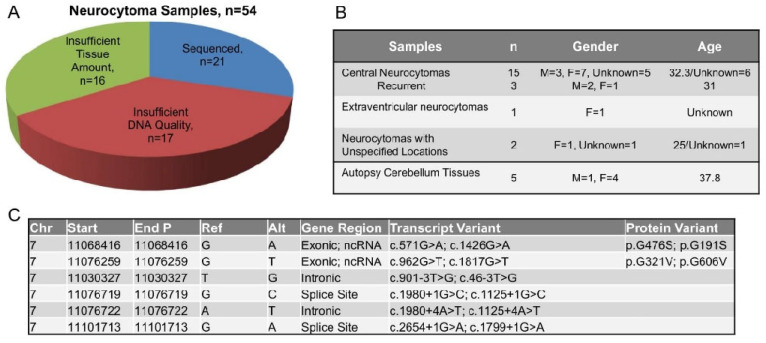
Identification of homeodomain finger protein 14 (PHF14) mutations in neurocytomas by whole exome sequencing. (**A**) Neurocytoma formalin-fixed, paraffin-embedded (FFPE) and frozen tissues (*n* = 54 total) were retrospectively collected from 11 centers across United States and Canada. Genomic DNA was extracted from 38 neurocytoma tissues with tumor area >95% and diameter >5 mm and was of suitable quality for whole exome sequencing in 21 neurocytoma tumors (FFPE samples *n* = 17, frozen samples *n* = 4). (**B**) Demographics and subtypes of 21 sequenced neurocytoma samples and 5 normal cerebellar controls. (**C**) Summary of the six PHF14 mutations detected by WES in three neurocytoma samples compared.

**Figure 2 biomedicines-10-02842-f002:**
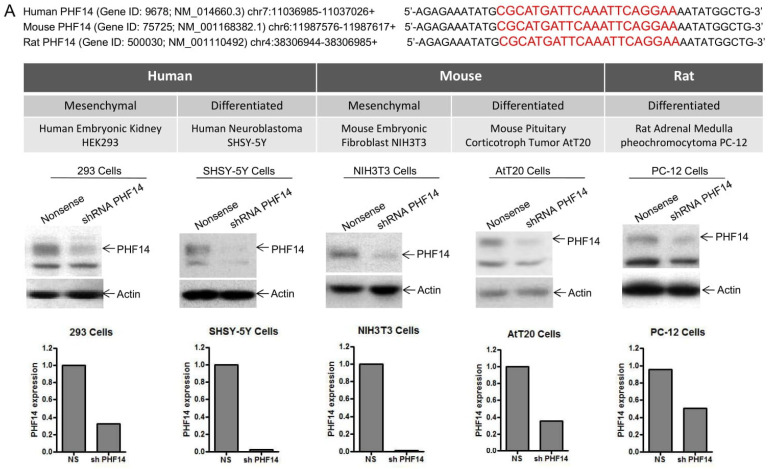
PHF14 Depletion Enhances Cell Proliferation and Anchorage Independent Cell Growth. (**A**) A conserved 19-bp region in the human, mouse and rat PHF14 gene in Exon 10 (CGC ATG ATT CAA ATT CAG GAA) was selected as shRNA target to knockdown PHF14 expression. Five cell lines originating from human, mouse and rat were used to evaluate the biological effect of PHF14 depletion. PHF14 knockdown stable transfectants were established by puromycin selection, and the knockdown efficiency was determined by Western Blot using anti-PHF14 antibody. The densitometric analyses of the protein bands vs. the individual loading controls were shown under individual blot using the ImageQuant 5.2 software (GE Healthcare, Pittsburgh, PA). (**B**) Cell proliferation rate was enhanced in shRNA PHF4 knockdown transfectants compared to Nonsense controls as measured by CellTiter-Glo^®^ luminescent cell viability assay in a variety of cell lines. (**C**) Soft agar assay demonstrated that PHF14 knockdown induced an increase in colony size in human neuroblastoma SHSY-5Y cells. (**D**) A single guiding RNA (sgRNA) targeting a 20-bp region (TGG ATC GCA GCT CCA AGA GG) in PHF14 Exon 1 was designed for CRISPR-Cas9 mediated genetic editing. The knockout efficiency was confirmed by Western Blot. PHF14 knockout in human neuroblastoma SHSY-5Y cells promoted cell proliferation as measured by CellTiter-Glo^®^ luminescent cell viability assay. PHF14 knockout increased colony formation in soft agar assay. The results shown were representative of three independent experiments. * *p* < 0.05; ** *p* < 0.01; *** *p* < 0.001.

**Figure 3 biomedicines-10-02842-f003:**
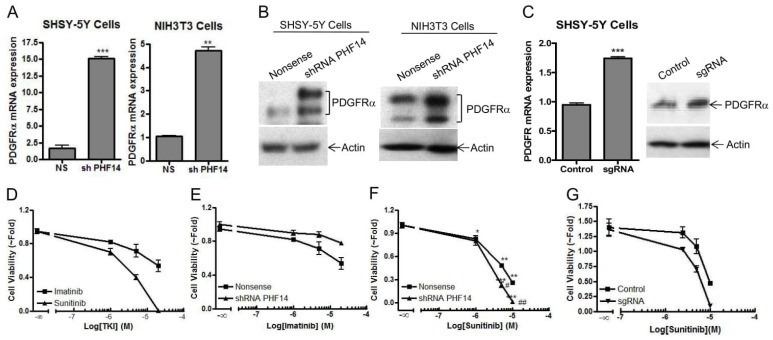
PHF14 Depletion Elevates platelet derived growth factor receptor-alpha (PDGFRα) Expression and Increases Responsivity to the PDGFR Inhibitor Sunitinib. (**A**–**B**) ShRNA-directed PHF14 knockdown increased PDGFRα expression at both mRNA (**A**) and protein (**B**) levels by Real Time PCR (**A**) and Western Blot (**B**) in human neuroblastoma SHSY-5Y cells and mouse embryonic fibroblasts NIH3T3 cells. (**C**) Increased PDGFRα expression at both protein and mRNA levels was confirmed in PHF14 knockout human neuroblastoma SHSY-5Y cells. (**D**) The anti-proliferation effects of PDGFR inhibitors, Imatinib and Sunitinib, were evaluated in SHSY-5Y cells by CellTiter-Glo^®^ luminescent cell viability assay. (**E** and **F**) PHF4 knockdown sensitized the anti-proliferation effect of PDGFR inhibitor Sunitinib (**F**), but not Imatinib (**E**) in human neuroblastoma SHSY-5Y cells as analyzed by CellTiter-Glo^®^ luminescent cell viability assay. (**G**) The sensitization to Sunitinib treatment was reproduced in PHF14 knockout SHSY-5Y cells. The results shown were representative of three independent experiments. * *p* < 0.05; ** *p* < 0.01; *** *p* < 0.005, compared with media control. # *p* < 0.05; ## *p* < 0.01 compared with Nonsense control.

**Figure 4 biomedicines-10-02842-f004:**
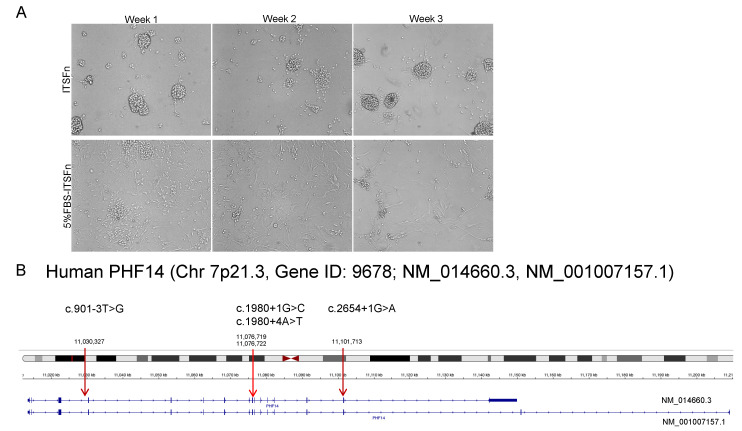
PHF14 Inhibits Cell Growth in Neurocytoma Primary Culture. (**A**) Typical morphologic appearance of neurocytoma primary cultures in serum free, insulin, transferrin, selenium and fibronectin (ITSFn) media (Top Panel) and serum supplemented ITFSn media (Bottom Panel) for 3 weeks to compare cell morphology changes. (**B**) Detection of four splicing loss mutations of PHF14 in the neurocytoma primary cultures by whole exome sequencing (WES). (**C**) Comparison of the transfection efficiency of Lipofectamine 2000 and transduction efficiency of lentivirus by introducing GFP-expressing vector. Wild type PHF14 was introduced into neurocytoma primary culture using lentivirus. The transduction efficiency was evaluated by Real Time PCR to detect PHF14 overexpression. Cell viability was detected by CellTiter-Glo^®^ luminescent cell viability assay. (**D**) Neurocytoma primary culture was treated with Sunitinib, MEK-162 and Metformin. Cell viability was detected by CellTiter-Glo^®^ luminescent cell viability assay. Data shown are representative of at least three independently conducted experiments. Bars indicate the mean ± standard error of the mean of triplicate tests. * *p* < 0.05; *** *p* < 0.005.

**Figure 5 biomedicines-10-02842-f005:**
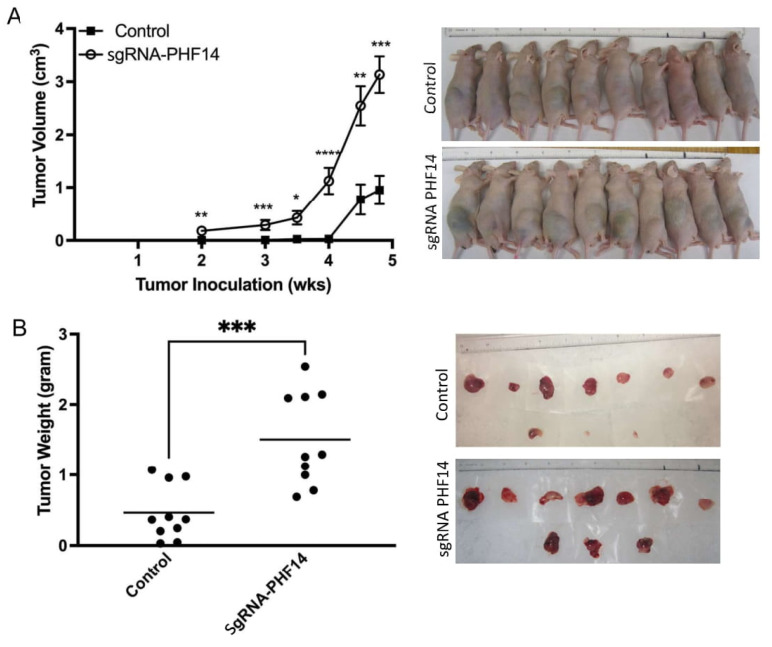
Evaluation of the Tumor Inhibitory Effect of PHF14 and Sunitinib Using in vivo Xenograft Animal Model. (**A**,**B**) Comparison of tumor development in PHF14 knockout cells (*n* = 10) and control neuroblastoma cells (*n* = 10) in an in vivo xenograft model of neuroblastoma. (**C**,**D**) To determine the effects of Sunitinib treatment on tumor growth in vivo, we administrated Sunitinib (dissolved in 0.5% CMC) by oral gavage following SHSY-5Y cells inoculation. The tumor growth was monitored and measured daily, and the animals were euthanized on day 21 after drug treatment due to deteriorating animal health condition. Tumors were excised, and weighed. Data were analyzed by two-tailed unpaired *t*-test, * *p* < 0.05; ** *p* < 0.01; *** *p* < 0.005; **** *p* < 0.0001.

## Data Availability

Please contact corresponding author for data accessibility.

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
