# Peer review of "Whole Exome Sequencing Identifies PHF14 Mutations in Neurocytoma and Predicts Responsivity to the PDGFR Inhibitor Sunitinib"

_biomedicines, 2022, doi:10.3390/biomedicines10112842_

Round 1
Reviewer 1 Report
In this manuscript the investigators performed WES of 21 neurocytomas (a rare low-grade brain tumors) and identified 6 somatic mutations in PHF14 in 3 of the tumors, “two were missense mutations and 4 caused splicing site losses”. Furthermore, they showed that knocking down PHF14 expression accelerated cell proliferation and colony formation and advanced in vivo tumor growth, suggesting that PHF14 is a tumot suppressor. Inspired by the 2012 publication of Kitagawa et al on mesenchymal cells, they examine PDGFR-α and showed that knockdown of PHF14 increased in PDGFR-α expression and increased sensitivity to sunitinib.
The experiments were carefully performremd and the reasults are statistically significant.
However, there are several issues that need to be addressed.
1) The main problem is that PDGFR-α expression was examined in 5 knockdown cells (only one established human neurocytoma SHSY-5Y that does not have the PHF14 mutation) but was not demonstrated in neurocytoma tumor tissues with and without mutations.
2) The authors imply that neurocytoma primary cutlure with PHF14 mutations overexpress PDGFRa because they are inhibited by sunitinib (Figure 4), but they don’t provide Western blot analysis with anti-PDGFR-α antibodies (before and after transfection with PHF14) to validate this point. Sunitinib is a multi-targeted receptor tyrosine kinase (RTK) inhibitor, including PDGFR-α, vascular endothelial growth factor receptors (VEGFRs), and c-KIT.
3) The investigators list 6 mutations in 3 specimens but do not indicate the number of mutations/specimens, and the type of mutation in a specific specimen.
4) the manuscript is written in a highly repetitous manner.
tumors. 36
resulting in 37
PHF14 dysfunction. Employing shRNA-mediated knockdown and CRISPR/Cas9-based knockout 38 approaches, we demonstrated that loss of PHF14 increased proliferation and colony formation in 39 five different human, mouse and rat mesenchymal and differentiated cell lines. Additionally, we 40 demonstrated that PHF14 depletion resulted in upregulation of PDGFR mRNA and protein in 41 neuroblastoma SHSY-5Y cells and led to increased sensitivity to treatment with the PDGFR inhibitor 42
Sunitinib. Furthermore, in a neurocytoma primary culture harboring splicing loss PHF14 mutations, 43
overexpression of wild-type PHF14 and sunitinib treatment inhibited cell proliferation. Nude mice, 44 inoculated with PHF14 knockout SHSY-5Y cells developed earlier and larger tumors
Author Response
October 12, 2022
Pawel Botwina, M.Sc.
Assistant Editor
MDPI
Biomedicines Editorial Office
Dear Mr. Botwina
Thank-you for the reviewer comments regarding our manuscript ID: biomedicines-1920735, entitled “Whole Exome Sequencing Identifies PHF14 Mutations in Neurocytoma and Predicts Responsivity to the PDGFR Inhibitor Sunitinib”. Below, we provide a point-by-point response to reviewers’ comments, and have amended the enclosed manuscript in the denoted sections to reflect these revisions.
Reviewer 1:
We are pleased reviewer 1 thought our experiments were carefully performed and the results are statistically significant.
Comment 1: "The main problem is that PDGFR-α expression was examined in 5 knockdown cells (only one established human neurocytoma SHSY-5Y that does not have the PHF14 mutation) but was not demonstrated in neurocytoma tumor tissues with and without mutations”.
Response 1: We agree that this is an important study but given the rarity of neurocytoma, we currently have insufficient specimens to examine PDGFR-alpha expression at present. We are actively collaborating with additional centers to collect more samples for validation.
Comment 2: “The authors imply that neurocytoma primary culture with PHF14 mutations overexpress PDGFRa because they are inhibited by sunitinib (Figure 4), but they don’t provide Western blot analysis with anti-PDGFR-α antibodies (before and after transfection with PHF14) to validate this point. Sunitinib is a multi-targeted receptor tyrosine kinase (RTK) inhibitor, including PDGFR-α, vascular endothelial growth factor receptors (VEGFRs), and c-KIT”.
Response 2: Analysis of whole transcriptome (Supp Fig. 1 and Supp Table 1) and western blot (Fig. 3B & C) in PHF14 knockdown human (SHY5-5Y) and murine (NIH3T3) cells demonstrated increased PDGFR- α expression (Page 6, 13 and 19). Due to extremely limited primary neurocytoma samples, we were unable to confirm these findings in primary neurocytoma.
Comment 3: “The investigators list 6 mutations in 3 specimens but do not indicate the number of mutations/specimens, and the type of mutation in a specific specimen”.
Response 3: A clear description of the mutations in individual specimens is now provided, Fig.1C (Page 5).
Comment 4: “The manuscript is written in a highly repetitious manner”.
Response 4: Thank-you. The manuscript has been revised to avoid repetition (Page 3 and 13).
Reviewer 2:
We appreciate that reviewer 2 thought that our study was novel and worthy of publication.
Comment 1: “PHF14 knockout/down may lead to enormous downstream alterations, why choose PDGFRa?”
Response 1: We agree that PHF14 knockdown may lead to multiple downstream effects. We chose to focus on PDGFR- α initially as a prior report had noted increased PDGFR-alpha in a biliary duct carcinoma harboring a PHF14 mutation (Akazawa et al., 2013) and our own transcriptome analysis noted increased PDGFR-alpha (> 1.5-fold) in human cell line following PHF14 knockdown (Supplementary Figure 1 and Table 1). This is more fully discussed in the results section (Page 6, 13 and 19).
Comment 2: “I am not sure whether we should call PHF14 "tumor suppressor gene". All evidences indicated that loss of PHF14 identified a severe type of neurocytoma which may be efficiently treated by PHDGFa inhibitor. It seems PHF14 serves as an alternative pathway for certain cancerous behavior”.
Response 2: We agree with this comment and have modified our wording (Page 2, 4, 6, 10 and 11).
Comment 3: “Based on Figure 5, sunitinib works fine for both control and PHF14 knockdown cells, although they are slightly different in sensitivity of response. Is this caused by endogenous PHF14 or PDGFa expression in SHSY-5Y?”
Response 3: Yes, we believe the response in the wild-type cells is reflective of endogenous PDGFRa expression although we cannot discount sunitinib actions via its other targets as noted by reviewer 1.
We hope that our revised manuscript is now suitable for publication in the section of Neurobiology and Neurologic Disease in Special Issue of 10th Anniversary of Biomedicines—Novel Targets for Cranial Tumors. Thank you for your consideration.
Yours sincerely
Dr. Anthony Heaney
Reference:
Akazawa, T., Yasui, K., Gen, Y., Yamada, N., Tomie, A., Dohi, O., Mitsuyoshi, H., Yagi, N., Itoh, Y., & Naito, Y. (2013). Aberrant expression of the PHF14 gene in biliary tract cancer cells. Oncology letters, 5(6), 1849-1853.

Reviewer 2 Report
The manuscript by Zhang et al., explored and investigated the role of PHF14 in neurocytoma, and identified its relevance to PDGFRa. Overall, the study is novel and worthy publishing, the authors did a lot of work, and I only have three rational concerns:
1) PHF14 knockout/down may lead to enormous downstream alterations, why choose PDGFRa?
2) I am not sure whether we should call PHF14 "tumor suppressor gene". All evidences indicated that loss of PHF14 identified a severe type of neurocytoma which may be efficiently treated by PHDGFa inhibitor. It seems PHF14 serves as an alternative pathway for certain cancerous behavior.
3) Based on Figure 5, sunitinib works fine for both control and PHF14 knockdown cells, although they are slightly different in sensitivity of response. Is this caused by endogenous PHF14 or PHDGFa expression in SHSY-5Y?
Author Response

(The authors gave the same response as above.)

Round 2
Reviewer 1 Report
Need to delete marked with " ": Herein we describe mutations in "the tumor suppressor gene", PHF14 (indicate the gene in italic) "mutation" in a series of neurocytomas.